# Effects in Israel of Arab and Jewish Ethnicity on Intracerebral Hemorrhage

**DOI:** 10.3390/jcm11082117

**Published:** 2022-04-11

**Authors:** Naaem Simaan, Andrei Filioglo, José E. Cohen, Yonatan Lorberboum, Ronen R. Leker, Asaf Honig

**Affiliations:** 1Department of Neurology, Ziv Medical Center, Safed 13100, Israel; naaem.simaan@gmail.com; 2Azrieli Faculty of Medicine, Bar Ilan University, Safed 13115, Israel; 3Department of Neurology, Hadassah-Hebrew University Medical Center, Jerusalem 91120, Israel; afilioglo@gmail.com (A.F.); yonilorby@gmail.com (Y.L.); asaf.honig2@gmail.com (A.H.); 4Department of Neurosurgery, Hadassah-Hebrew University Medical Center, Jerusalem 91120, Israel; jcohenns@yahoo.com

**Keywords:** intracerebral hemorrhage, Arab ethnicity, Jewish ethnicity, hypertensive arteriopathy

## Abstract

Intracerebral hemorrhages (ICH) characteristics reportedly differ between different ethnic groups. We aimed to compare the characteristics of Jewish and Arab ICH patients in Israel. Consecutive patients with primary ICH were included in a prospective institutional database. Demographics, vascular risk factors, clinical and radiological parameters were compared between Arab and Jewish ICH patients residing in Jerusalem. The study included 455 patients (311 Jews). Arab patients were younger (66.1 ± 13.4 vs. 72.2 ± 12.2 years, *p* < 0.001) and had higher rates of diabetes (60% vs. 29%, *p* < 0.001) and smoking (26% vs. 11%, *p* < 0.001). Arab patients had higher rates of deep ICH (74% vs. 62%, *p* = 0.01) and lower rates of lobar ICH (18% vs. 31%, *p* = 0.003). In a sub-analysis of deep ICH patients only, Arab patients were younger (64.3 ± 12.9 vs. 71.4 ± 11.8 years, *p* < 0.001) and less frequently male (56% vs. 68%, *p* = 0.042), with higher rates of diabetes (61% vs. 35%, *p* < 0.001) and smoking (31% vs. 14%, *p* < 0.001). In conclusion, the two ethnic populations in Israel differ in the causes and attributes of ICH. Heavy smoking and poorly controlled diabetes are commonly associated with deep ICH in the Arab population and may offer specific targets for secondary prevention in this population.

## 1. Introduction

Intracerebral hemorrhage (ICH) accounts for 10–20% of all strokes [1,2]. ICH is known for its devastating outcomes, including its higher morbidity and mortality rates than ischemic stroke, with 50% of the deaths occurring during the first month [3,4].

Previous studies focusing on the incidence of ICH in different ethnic groups [5,6,7,8] demonstrated that in the USA, the highest ICH rates were observed in Asians, followed by African Americans, and Caucasians [5,9,10,11,12,13].

In Israel, ICH incidence and patient characteristics in Jewish and Arab patients has not been compared. The current study aimed to explore potential differences in vascular risk factors, etiologies, clinical presentations, and outcomes of ICH between these two ethnic groups.

## 2. Materials and Methods

Consecutive patients with primary ICH were included in an ongoing prospective all-inclusive institutional database. The institutional review board (Hadassah Medical Organization) authorized the anonymous inclusion of ICH patients into the consecutive database with a waiver of the requirement for informed consent (HMO-437-20).

The database includes epidemiological, clinical, and radiological information. Ethnicity data, trichotomizing patients into Jewish, Arab, and other ethnic groups were taken from each patient’s identification card. Only Jewish and Arab patients residing in Israel were included in the current study, allowing comparisons to be made between two populations receiving similar health care in the same system, and living in a similar climate [14]. Vital status on day 90 (survival vs. death) was acquired through online connection between the electronic medical file and the Israeli Ministry of Interior office.

In the current analysis, we included patients with ICH admitted between 1 January 2009 and 1 October 2020. Patients with traumatic ICH were excluded, as were those with hemorrhagic transformations of ischemic infarcts and patients with hemorrhages in primary brain tumors or metastases.

The diagnosis of ICH was established according to clinical findings and a baseline noncontrast CT scan that showed the hemorrhage. In addition, most patients had undergone CT angiography (CTA) to identify the presence of spot signs and for ruling out underlying vascular etiology. Patients routinely had a follow up noncontrast CT 24 h after admission. In every case where the ICH presentation was atypical, further radiologic evaluation using CT angiography, angiography, and MR imaging were performed. In addition, follow-up MRI was recommended to all patients after full hematoma resolution to reveal any underlying structural etiology as well as the presence of cerebral microbleeds.

Experienced stroke neurologists, blinded to the clinical scenario, reviewed all imaging studies. ICH location was determined based on the first noncontrast CT and was categorized as lobar, basal ganglia-thalamus, brainstem, or cerebellum. Brainstem and basal ganglia hemorrhages were collectively defined as deep ICH. Hematoma volumes and the presence/absence of intraventricular extension were assessed on follow-up CT scans.

Demographics and vascular risk factors such as hypertension, diabetes, hypercholesterolemia, and chronic renal failure were taken from both the hospital admission files. Heavy smoking was defined as smoking more than 20 cigarettes per day for more than ten years.

Data analysis was performed with the statistical software SPSS version 24 (IBM USA). The two-sample *t*-test was applied for testing differences between the study groups for quantitative parameters. Pearson’s chi-squared and Fisher’s exact tests were applied to test differences between the ethnic groups and outcome groups for the categorical parameters. A *p*-value of 5% or less was considered statistically significant.

## 3. Results

A total of 455 patients (311; (68%), Jewish and 144; (32%), Arab) were included in the study. Arab ICH patients were younger (66.1 ± 13.4 vs. 72.2 ± 12.2 years, *p* < 0.001) and had higher rates of diabetes (60% vs. 29%, *p* < 0.001) and smoking (26% vs. 11%, *p* < 0.001, Table 1). ICH location was significantly different between the ethnic groups (*p* = 0.013); Arab ICH patients presented more often with deep ICH (74% vs. 62%, *p* = 0.01), and Jewish ICH patients presenting more often with lobar ICH (31% vs. 18%, *p* = 0.003). All other parameters tested, including the frequency of high cholesterol, chronic renal failure, dual antiplatelet therapy (DAPT), anticoagulation therapy, platelets, international normalized ratio (INR), creatinine, estimated glomerular filtration rate (eGFR), and vital status were comparable between groups.

In comparisons between Arab and Jewish patients with deep ICH (Table 2), the same trends were found. Arab patients were younger (64.3 ± 12.9 vs. 71.4 ± 11.8 years, *p* < 0.001) and presented with higher rates of diabetes (61% vs. 35%, *p* < 0.001) and heavy smoking (31% vs. 14%, *p* < 0.001). A higher proportion of Arab patients with deep ICH were female (44% vs. 32%, *p* = 0.042). Platelet counts were significantly higher in the Arab patient population (240 ± 85 vs. 208 ± 64, *p* = 0.006). All other parameters tested were comparable between groups.

When comparing between Jewish and Arab lobar ICH patient characteristics (Table 3), the only significant difference observed between groups was the higher rates of diabetes found in Arab patients (52% vs. 19%, *p* = 0.001).

Deep vs. lobar ICH patients were younger (68.9 ± 12.7 vs. 72.9 ± 13.9, *p* = 0.004) more often males (64% vs. 45%, *p* < 0.001), of Arab ethnicity (36% vs. 21%, *p* = 0.003) and previously diagnosed with chronic hypertension (83% vs. 65%, *p* < 0.001), diabetes (44% vs. 25%, *p* < 0.001) and stroke (24% vs. 15%, *p* = 0.036). In the multivariate, analysis independent variables associated with deep ICH were male sex (OR 2.05, 95% CI 1.3–3.24, *p* = 0.002), chronic hypertension (OR 2.05, 95% CI 1.21–3.45, *p* = 0.007) and Arab ethnicity (OR 1.74, 95% CI 1–3.04, *p* = 0.05). Age (OR 0.986 per year increase, 95% CI 0.97-1.004, *p* = 0.12), diabetes (OR 1.4, 95% CI 0.84-2.36, *p* = 0.196) and history of previous stroke (OR 1.61, 95% CI 0.87–3, *p* = 0.132) were not found to be independently significant.

## 4. Discussion

The current study found that that Arab ICH patients are significantly younger, with the age difference being most pronounced in the deep ICH population. This is not surprising, as the Arab patients in our cohort more frequently had risk factors such as diabetes and smoking, which play a significant role in the pathogenesis of cerebral small vessel arteriopathy and consequently deep ICH [15].

Of note, the higher prevalence of smoking and diabetes in the Arab population was also seen in both the Israeli National Acute Stroke Survey and in a study in northern Israel [16,17,18]. In both studies, lacunar strokes were more common in the Arab population and the Arab patients were significantly younger, corroborating our data.

We found that Arab ICH patients showed a trend to suffer more often from hypertension. However, it is possible that the diagnosis of hypertension was delayed in this population due to lower rates of family medicine visits, which may have also led to less adequate blood pressure control [19]. Our findings also raise the question of whether there is genetic susceptibility that may leave patients more prone to developing hypertensive arteriopathy in the Arab population.

Data analysis from the ERICH study (Ethnic/Racial Variations of Intracerebral Hemorrhage), a prospective, multicenter, case–control study, found that while hypertension contributed to both lobar and deep ICH in all ethnicities, both treated and untreated hypertension were stronger predictors of ICH in the African American and Hispanic populations in comparison to the Caucasian population [20]. Similarly, in an MRI subanalysis of the Massachusetts General Hospital ICH study and the ERICH study, cerebrovascular disease burden was quantified using validated hypertension arteriopathy-specific scores [21]. African American and Hispanic survivors of hypertensive arteriopathy-related ICH had a higher hypertension arteriopathy burden, resulting in an increased risk of ICH recurrence [21].

Moreover, a more detailed assessment of the role of genetics and ethnicity may play a role in identifying individuals more susceptible to ICH than lacunar stroke [12,22,23]. Reports from East Asian countries indicate a higher incidence of spontaneous ICH compared to those seen in Western countries [12]. Over the past twenty years, the incidence of ICH in China has more than doubled among middle-aged men [24]. Increased rates of conventional vascular risk factors in China, including obesity and hypertension, are associated with an overall increase in all stroke types, but have still been associated with higher rates of ICH than ischemic strokes. For instance, the mean age of first symptomatic ICH in men decreased by 0.56 years annually compared to an annual decrease of only 0.22 years for ischemic stroke [25].

Our study found that rates of diabetes were twice as high in the Arab ICH patients. Diabetes has been suggested to play a role in ICH in many studies, and an increased risk of ICH has been directly associated with diabetes duration [26]. Interestingly, this frequency was found in a subanalysis of both deep and lobar ICH, suggesting that diabetes has a role as a contributing factor to ICH related to both hypertensive angiopathy and cerebral amyloid angiopathy.

Another characteristic differentiating Arab and Jewish ICH patients was heavy smoking. The contributing role of heavy smoking has been previously described in population studies, as well as in the INTERSTROKE study, an international case–control study of 6000 individuals, which showed that heavy smoking is a major risk factor for ICH [27,28]. Moreover, recent publications have suggested that heavy smoking increases the risk of spontaneous primary ICH and perihematomal edema and is consequently associated with worse outcomes [29].

Finally, we found that male sex, chronic hypertension, and Arab ethnicity were independently associated with deep rather than lobar ICH. The independent impact of ethnicity on the tendency to experience deep ICH is very intriguing and warrants more pathophysiologic and genetic research in addition to more aggressive primary and secondary prevention policies. Of note, the independent role of Arab ethnicity was of borderline significance (*p* = 0.05) and may have been the result of factors we did not encounter in our analysis.

## 5. Conclusions

The increased rate of deep ICH could be attributed to heavy smoking, diabetes, and possibly poorly controlled hypertension in the Arab Israeli population. These modifiable risk factors should be the target of extensive preventive campaigns aiming to lower the risk of ICH in this population.

## Figures and Tables

**Table 1 jcm-11-02117-t001:** Comparison of ICH characteristics for Arab and Jewish patients.

Characteristics	Arab N = 107	Jewish N = 311	*p*
Age, mean (SD)	66.1 (13.4)	72.2 (12.2)	**<0.001**
Gender male (%)	78 (54)	183 (59)	0.348
Hypertension (%)	120 (84)	236 (76)	0.053
Elevated cholesterol (%)	54 (52)	36 (43)	0.193
Smoking (%)	37 (26)	35 (11)	**<0.001**
Diabetes (%)	85 (60)	91 (29)	**<0.001**
Chronic renal failure (%)	18 (17)	42 (19)	0.710
Prior stroke (%)	38 (28)	61 (20)	0.081
DAPT (%)	51 (39)	121 (42)	0.571
Anticoagulants (%)	24 (19)	42 (15)	0.187
ICH location (%)			**0.013**
Deep	107 (74)	193 (62)	
Lobar	26 (18)	97 (31)	
Cerebellar	11 (8)	21 (7)	
Ventricular extension (%)	66 (52)	110 (47)	0.368
ICH volume, mean (SD)	58.5 (204.9)	60.9 (91.1)	0.892
Platelets, mean (SD)	236 (86)	219 (83)	0.107
INR, mean (SD)	1.3 (0.8)	1.4 (0.9)	0.460
Creatinine, mean (SD)	119 (116)	105 (110)	0.235
EGFR, mean (SD)	74.5 (28.5)	70.2 (23.9)	0.139
90-day mortality (%)	58 (40)	134 (43)	0.670

Values represent number of patients unless otherwise stated. ICH = intracerebral hemorrhage; DAPT = dual antiplatelet therapy; INR = international normalized ratio; EGFR = estimated glomerular filtration rate. Bold numbers signify statistically significant *p* values.

**Table 2 jcm-11-02117-t002:** Comparison of ICH Arab and Jewish patients with deep ICH.

Characteristics	Arab N = 107	Jewish N = 311	*p*
Age, mean (SD)	64.3 (12.9)	71.4 (11.8)	**<0.001**
Gender male (%)	60 (56)	131 (68)	**0.042**
Hypertension (%)	92 (87)	157 (81)	0.227
Elevated cholesterol (%)	36 (49)	26 (42)	0.434
Smoking (%)	33 (31)	26 (14)	**<0.001**
Diabetes (%)	65 (61)	67 (35)	**<0.001**
Chronic renal failure (%)	14 (19)	26 (20)	0.817
Prior stroke (%)	28 (28)	42 (23)	0.344
DAPT (%)	41 (43)	72 (42)	0.862
Anticoagulants (%)	14 (15)	24 (15)	0.841
Ventricular involvement (%)	51 (56)	75 (52)	0.553
ICH volume, mean (SD)	31.9 (44.3)	38.9 (52.3)	0.428
Platelets, mean (SD)	240 (85)	208 (64)	**0.006**
INR, mean (SD)	1.2 (0.6)	1.4 (0.8)	0.177
Creatinine, mean (SD)	123 (125)	110 (129)	0.408
EGFR, mean (SD)	73.7 (28.6)	70.4 (23.7)	0.326

ICH = intracerebral hemorrhage; DAPT = dual antiplatelet therapy; INR = international normalized ratio; EGFR = estimated glomerular filtration rate. Bold numbers signify statistically significant *p* values.

**Table 3 jcm-11-02117-t003:** Characteristics of Arab and Jewish lobar ICH patients.

Characteristics	Arab N = 107	Jewish N = 311	*p*
Age, mean (SD)	69.9 (15.5)	73.6 (13.5)	0.235
Gender male (%)	14 (54)	41 (42)	0.292
Hypertension (%)	18 (69)	62 (64)	0.614
Cholesterol (%)	11 (61)	51 (53)	0.653
Smoking (%)	4 (16)	8 (8)	0.246
Diabetes (%)	13 (52)	18 (19)	**0.001**
Chronic renal failure (%)	4 (18)	11 (14)	0.587
Prior stroke (%)	5 (20)	13 (14)	0.419
Dementia (%)	1 (5)	16 (20)	0.092
DAPT (%)	6 (24)	36 (38)	0.195
Anticoagulants (%)	7 (29)	14 (15)	0.108
Ventricular involvement (%)	11 (44)	25 (34)	0.383
ICH volume, mean (SD)	167.7 (440)	112.2 (129.1)	0.393
Platelets, mean (SD)	204 (79)	238 (116)	0.260
INR, mean (SD)	1.7 (1.3)	1.5 (1.3)	0.583
Creatinine, mean (SD)	113 (105)	92 (59)	0.184
EGFR, mean (SD)	79.5 (32.2)	71.6 (23.8)	0.236

ICH = intracerebral hemorrhage; DAPT = dual antiplatelet therapy; INR = international normalized ratio; EGFR = estimated glomerular filtration rate. Bold numbers signify statistically significant *p* values.

## Data Availability

Questions regarding data should be addressed to the corresponding author.

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
