# Peer review of "Effects in Israel of Arab and Jewish Ethnicity on Intracerebral Hemorrhage"

_jcm, 2022, doi:10.3390/jcm11082117_

Round 1

Reviewer 1 Report

The work is interesting but it does not show the difference between different ethnic groups but between different populations. Which differ in terms of diabetes, blood pressure and smoking, exactly as the authors concluded. To identify ethnicity difference we should have comparable groups. Is it possible to use propensity score usino different age groups? I would add analyses for groups of equal age or similar risk factors to assess the effect of ethnicity on ICH.

Author Response

Reviewer 1 has very accurately pointed out that our work does not show the difference between different ethnic groups but between different populations. 

In order to show the effect of ethnicity on ICH we have performed a multivariate analysis for predictors of deep intracerebral hemorrhage (ICH). As added to the last paragraph of the results section, we have found that Arab ethnicity was independantly associated with deep ICH.  

Reviewer 2 Report

Thank you for the opportunity to review this paper.

The authors avualute differences between Arab and Jewish patients with spontaneous cerebral hemorrhage and found more deeply located hemorrhages in Arabs, who also had significantly more often Diabetes and were more often smokers. Although the content may not be relevant to all readers on a day-to-day basis, the findings are interesting and draw attention to ethnic differences that are often ignored in the evaluation of different diseases. 

I have a few comments/questions:
Are there any data on the prevealence of diabetes and nicotine abuse in general in the Arab and Jewish populations of Israel? Say: do Arabs smoke more often and suffer from diabetes more often? 
Is it possible to speculate if these results only apply to Arabs living in Israel or can this be generalized to other countries?
What role do genetic factors play? Can you say how many patients especially with lobar hemorrhages have an underlying amyloid angiopathy?

Author Response

We thank Reviewer 2 for the following questions:

Are there any data on the prevealence of diabetes and nicotine abuse in general in the Arab and Jewish populations of Israel? Say: do Arabs smoke more often and suffer from diabetes more often? 

Unfortunately, according to data given by the Israeli ministry of health, the rates of smoking in the Arab Israeli population is higher than that in the Jewish Israeli population (24.4% vs 19.1%, p<0.001). Moreover, in our work of patients who experienced ICH in our tertiary care center, the differences between the two ethnicities were even more pronounced. 

Is it possible to speculate if these results only apply to Arabs living in Israel or can this be generalized to other countries?

We were being very careful with the generability of the results and preferred to present them and allow the readers to judge if they can be generalized.

What role do genetic factors play?

We have speculated in the discussion section that genetic tendency may play a role.

Can you say how many patients especially with lobar hemorrhages have an underlying amyloid angiopathy?

Unfortunately, as not all the patients in our cohort who suffered from lobar hemorrhage underwent MRI, we cannot provide that information. 

This manuscript is a resubmission of an earlier submission. The following is a list of the peer review reports and author responses from that submission.